# A pragmatic approach for producing theoretical syntheses in ecology

**Bruno Travassos-Britto**[1,2]*, **Renata Pardini**[2,3], **Charbel N. El-Hani**[2,4,5], **Paulo I. Prado**[1,2]

**1** Department of Ecology, University of São Paulo, São Paulo, São Paulo, Brazil, **2** National Institute of Science and Technology in Interdisciplinary and Transdisciplinary Studies in Ecology and Evolution, Brazil, **3** Department of Zoology, University of São Paulo, São Paulo, São Paulo, Brazil, **4** Department of General Biology, Federal University of Bahia, Salvador, Bahia, Brazil, **5** Centre for Social Studies, University of Coimbra, Coimbra, Portugal

* bruno.travassos@usp.br

**Data Availability Statement:** All relevant data are within the paper and supporting information files.

**Funding:** BTB was supported by a scholarship from CAPES (grant number 88882.315632/2019-01) during the time of this study. RP, CE and PP

## Abstract

It has been proposed that ecological theory develops in a pragmatic way. This implies that ecologists are free to decide what, from the knowledge available to them, they will use to build models and learn about phenomena. Because in fields that develop pragmatically knowledge generation is based on the decisions of individuals and not on a set of predefined axioms, the best way to produce theoretical synthesis in such fields is to assess what individuals are using to support scientific studies. Here, we present an approach for producing theoretical syntheses based on the propositions most frequently used to learn about a defined phenomenon. The approach consists of (i) defining a phenomenon of interest; (ii) defining a collective of scientists studying the phenomenon; (iii) surveying the scientific studies about the phenomenon published by this collective; (iv) identifying the most referred publications used in these studies; (v) identifying how the studies use the most referred publications to give support to their studies and learn about the phenomena; (vi) and from this, identifying general propositions on how the phenomenon is approached, viewed and described by the collective. We implemented the approach in a case study on the phenomenon of ecological succession, defining the collective as the scientists currently studying succession. We identified three propositions that synthesize the views of the defined collective about succession. The theoretical synthesis revealed that there is no clear division between "classical" and "contemporary" succession models, and that neutral models are being used to explain successional patterns alongside models based on niche assumptions. By implementing the pragmatic approach in a case study, we show that it can be successfully used to produce syntheses based on the actual activity of the scientific community studying the phenomenon. The connection between the resulting synthesis and research activity can be traced back through the methodological steps of the approach. This result can be used to understand how knowledge is being used in a field of study and can guide better informed decisions for future studies.

received research fellowships from CNPq (grant numbers 311051/2018-9, 303011/2017-3 and 310885/2017-5 respectively). CE was also funded by CAPES and UFBA for Senior Visiting Researcher Grant included in the CAPES-PRINT Program, which funds his stay in the Centre for Social Studies, University of Coimbra, Portugal (grant number 88887.465540/2019-00). CNPq (grant number 465767/2014-1) and CAPES (grant number 23038.000776/2017-54) supported INCT IN-TREE, which the current study is part of. CAPES website:https://www.gov.br/capes/pt-br CNPqwebsite:https://www.gov.br/cnpq/pt-br The funders had no role in study design, data collection and analysis, decision to publish, or preparation of the manuscript.

**Competing interests:** The authors have declared that no competing interests exist.

## Introduction

Ecology is missing a framework that helps organize the knowledge generated within this field. This science has undergone intense development for more than a century of history which has led to a great number of very useful models to learn about phenomena [1–4]. Organizing these models into a cognitively manageable framework could help prevent spurious debate and foster quicker identification of gaps of knowledge [3–5]. This clarity leads to a more efficient way to generate knowledge.

Most previous attempts to delineate such a framework were based on expert opinion and assumed the possibility to identify a set of fundamental principles guiding model building and use in ecology [e.g. 5–7]. However, despite counting on the expertise of renowned authors in this field, these attempts have led to no clearly agreed fundamental principles unifying all ecological science [3–5]. One explanation for this is that ecologists seldom feel the need to use models that are conceptually unified when trying to learn about their phenomena of interest, a behaviour that is rather common in sciences that develop pragmatically [8, 9].

Under the pragmatic view, models are built freely by resorting to any knowledge available to the modeller such as previously proposed models, propositions, methods, and concepts. The final decision on which of those are or are not useful is made by the agents (in this case, practising ecologists) when carrying out their studies. As a consequence, the set of models about a phenomenon in a field of study is defined by the decisions of a collective of scientists trying to learn about the phenomenon, not by the deductive relations of a model with a set of fundamental principles [9]. In this scenario, finding a field of study where all models used are unified by a set of fundamental axioms would be an occasional occurrence, not a rule. Therefore, a framework based on the conceptual unification assumption would not be adequate to generate theoretical syntheses in all fields of study within a pragmatic science, as is the case of ecology [10].

Instead of using a framework that relies on the identification of a set of fundamental principles guiding model building and use, we propose an approach based on the actual use of models by scientists. Scientists report their research mostly in written articles where they make propositions about how the world is (*sensu* McGrath & Frank [11]). Propositions made in past studies are often referred to in subsequent studies to inform readers of which views about the world the research is based upon. Therefore, by accessing the article reporting a scientific study it is possible to trace which propositions are being used as the conceptual basis for that study and how they are being used [12]. The same applies to a collective of scientists studying a phenomenon. By accessing the articles describing the scientific studies of a collective, one can discover which and how propositions are being used to learn about the world from the point of view of that collective.

In the next section, we present a general approach that can be used to identify and describe the most referred propositions in scientific research about a certain phenomenon, mechanism or process within a field of study. Afterwards, we present the specific way in which we implemented this general approach as to identify the current conceptual bases of the studies about ecological succession and describe the specific methodological decisions we made to implement each step as well as the results obtained. These results are used to discuss in what ways our proposed approach to producing a theoretical synthesis is different from another study with the same aim but that uses the more traditional "expert opinion" approach. We end up presenting how syntheses produced by using our approach can fulfil the role of presenting and guiding knowledge in a field of study attributed to clearly defined theories.

## PATh–Pragmatic Approach to Theories

The notion that model validity corresponds to how much a model is used in a given context is based on the pragmatic view of scientific theories [8]. Therefore, we dubbed our approach Pragmatic Approach to Theories (PATh). We outlined the PATh as a set of steps that can be executed by adopting different methods. Here, we describe this set of steps by pointing out what each is supposed to achieve in a more general conceptual way. Afterwards, we will describe how we specifically implemented each step in a case study about ecological succession. The case study shows that it is possible to execute the conceptual approach in a more practical way, it also exemplifies the kind of information one can have by using the approach.

The approach we are proposing consists of the following steps (see also Fig 1):

I.  **Definition of the phenomenon of interest:** The goal is to define the object of the theory. In the natural sciences, theories are about natural phenomena. Defining the phenomenon of interest should specify which concepts and corresponding terms to look for in the scientific literature. The output of this step should be a set of terms that circumscribe which studies are regarded to be about the selected phenomenon.

II.  **Definition of the collective of scientists studying the phenomenon:** The goal is to define from whose point of view one wants to make a theoretical synthesis about the phenomenon. Defining the collective of scientists will give hints on where to look and how to filter all the activities aimed at learning about the phenomenon of interest. The output of this step should be a set of search parameters related to time and geographic scopes and

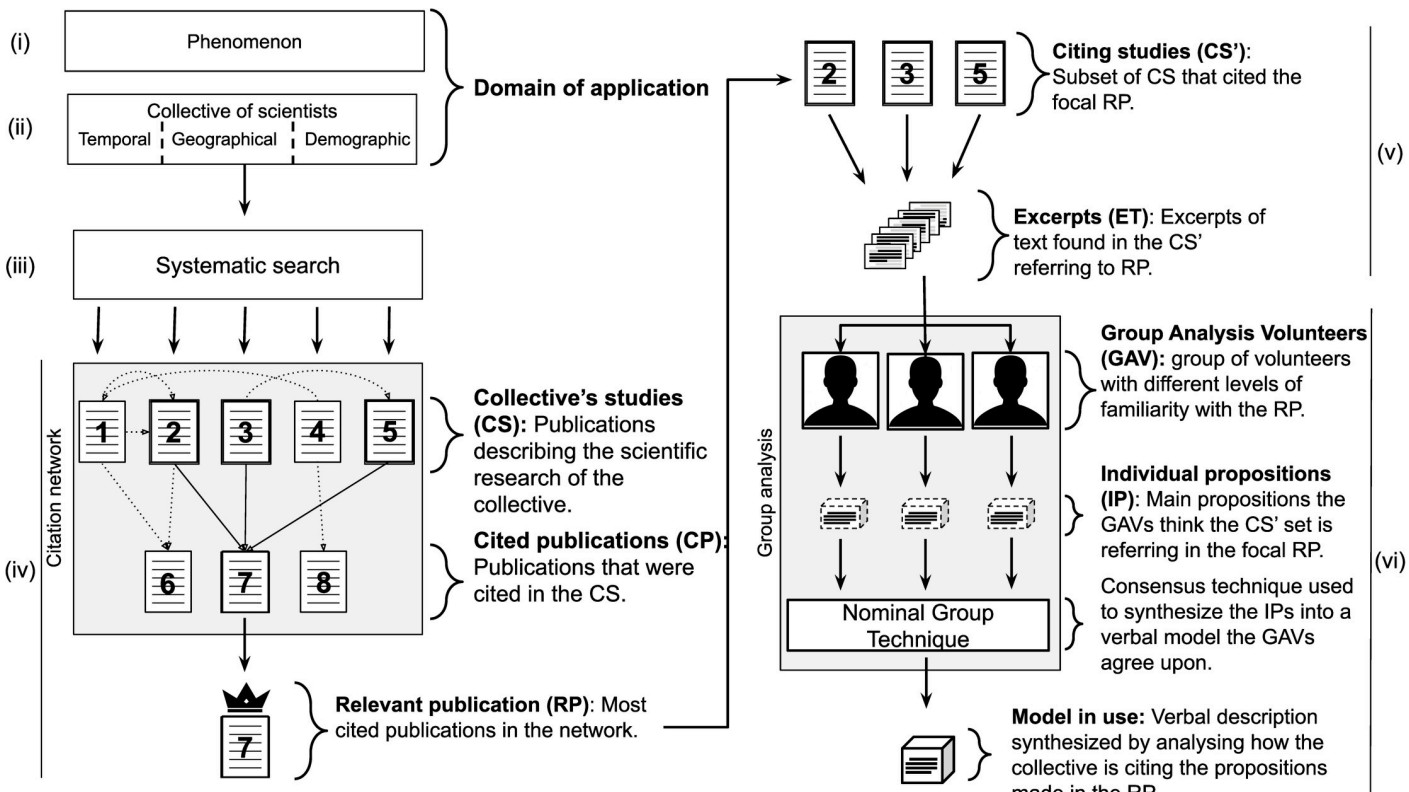

**Fig 1. PATh workflow.** Workflow depicting how the information obtained in each of the steps of the PATh (roman numerals) is used in the following step aiming to go from the definition of the domain of application to the final models in use. Note that the steps (v) and (vi) are carried out for each most referred publication in the set defined in step (iv).

demographic and academic profile of the focal collective of scientists studying the phenomenon of interest.

III. **Survey of the scientific activity:** The goal of this step is to take a picture of the activity of the focal collective investigating the phenomenon of study. This collective may present their activities in different ways. Currently, the most used way to report a scientific study is in academic journals. With the parameters resulting from the previous step, one will be able to conduct a search aimed at recovering the reports of the studies of the focal collective about the phenomenon of interest. The output of this step is a set of publications that should be a meaningful sample of the publications of the collective aimed at learning about the phenomenon.

IV. **Identification of relevant publications:** The goal of this step is to identify the publications that were most referred to due to their conceptual contribution to the scientific activity of the collective of scientists defined in step (iii). The output of this step should be a set of most referred publications among those identified in step (iii).

V. **Excerpting:** The goal of this step is to identify, among all the information contained in the most referred publications, what is effectively being used by the community in their scientific activity and how it is being cited. The result of this step is a set of excerpts of text containing the citation of the most referred publications as described in the studies carried out by the collective of scientists.

VI. **Content analysis of excerpts:** The goal is to rebuild self-consistent descriptions of the focal phenomena or other related phenomena based on the excerpts of text containing the citations of the most referred publications. Common statements among these excerpts are used to rebuild the self-consistent descriptions–referred here as "models in use". The result of this step is a set of statements that synthesize the main contributions of the relevant publications used by the collective to learn about the phenomenon.

The approach assumes that science works as a decentralized system of information exchange (*sensu* Von Bertalanffy [13]). This system has been described in many ways, e.g., as cycles of normal science followed by paradigm shifts [14], as heterogeneous networks of actors [15], or as distributed cognitive systems [16]. Our approach is agnostic on the details of the dynamics of scientific information exchange systems, provided that such dynamics includes the use of models (as defined above). It is worth noting that these are conceptual models rather than models of data. As a consequence syntheses produced using the PATh are conceptual and should be viewed as complementary instead of an alternative to meta-analysis or other approaches used to reach consensus across quantitative data.

The steps included in the PATh were thought to give access to the activities of the collective of scientists studying a phenomenon and, at the same time, deal with the issues brought about by assuming that the decentralized system of information exchange can be traced in the network of citations among publications. The first issue is the gradient in citations across publications in a network, requiring a cut-off criteria to circumscribe the most referred publication. The second issue is that the citation index alone does not necessarily reflect theoretical links between citing and cited publication [12, 15, 17]. Therefore, it is necessary to differentiate citations that reveal theoretical links from other types of citations. The third issue is that even when a publication is cited due to its theoretical relevance in a field of study, it is most likely that only a particular part of it is being considered relevant and not all the propositions contained in it. Therefore, it is necessary to identify what part of the relevant publications is actually being referred to by the studies being carried out by the collective.

The steps (iv), (v) and (vi) were designed to deal with these issues and in doing so they differentiate this approach from a simple analysis based on a systematic search. In these steps, one should define criteria to reach a finite set of relevant publications, to differentiate theoretically relevant citations from non-relevant ones, and to identify what the collective of scientists is actually referring to in the relevant publications.

It is important to note that the general approach described here can be implemented in different ways. In the following section, we provide an example of how the approach can be implemented. We specifically describe the methods adopted in each step, taking ecological succession as a case study. It is important to note that the methods we describe in the following section are not the PATh, but rather one way to carry out the approach. These methodological steps seemed as the best one to deal with our chosen domain of study, but this might not be true to other domains.

## A case study on ecological succession

### Defining the phenomenon of interest

To present a detailed example of how PATh can be implemented, we chose ecological succession as a case study. Ecological succession is one of the first phenomena studied in ecology, and had a central role in ecological theories [18]. Moreover, theoretical syntheses about the phenomenon of succession have been proposed in different moments in time and using different approaches [6, 7, 18, 19]; and it is still intensively studied. These characteristics of this field make it the ideal object for the first implementation of the PATh because there will be no shortage of material to compare our results with. Furthermore, regardless of its long history, the phenomenon is referred to by a single term ("succession" itself), with a small degree of ambiguity or polysemy, compared to other ecological concepts [18]. We realize that this choice leads to a very inclusive criterion about what is ecological succession, but despite that, it points to a well-circumscribed research field in ecology. Inclusive definitions are not problematic or uncommon in theoretical synthesis, as exemplified by Pickett, Meiners and Cadenasso's [19, p. 187] definition of succession, regarded by the authors themselves as very inclusive: "changes in structure or compositions of a group of organisms of different species at a site through time". Our approach is, however, robust to other meanings ascribed to the term, as we will see in the following section.

### Defining the collective of scientists

At this step, we chose to limit our case study to a collective of scientists currently publishing their results in venues that are part of a bibliographic database with broad coverage of English-written papers. What is considered as "current" is rather arbitrary and, therefore, we used a common option in literature surveys, which typically include a range of 10 years. We decided that the present decade (at the time of the analysis, 2007 to 2017) was an adequate scope of time, considering the relatively long time a new discovery about the world takes to change a field of study [20].

We did not restrict the geographical or demographic scopes to a specific group within this collective beyond the coverage provided by the database, and we did not take additional measures to control for publication or citation biases that may exist in the database (see [21]). Therefore, the results we obtained are as geographically and demographically biased as the publications found in the database used in the analysis.

### Surveying the scientific activities

This step was accomplished by searching all publications in the database in the past ten years (2007 to 2017) about the phenomenon of ecological succession. To do so we carried out a

systematic search in the ISI Web of Science™ database. This database was chosen because it has a set of tools for systematic search as well as output files that can be promptly used to make citation network analyses (see next section). We used the keyword "successi*" in the "Topic" field of the search engine of the database, which searches for the term in titles, abstracts, keywords and "keywords plus", and filtered the publications that were in the "Ecology" category. We filtered the search for publications dated from January 2007 to July 2017. This search resulted in 5,536 publications, which represent a sample of the documentation of the scientific activity related to studies on succession in ecology during the past ten years.

## Identifying relevant publications

To identify the most referred publications, we built a citation network including the 5,536 publications representing the studies of the focal collective plus all publications directly cited by these studies using the software CitNetEplorer™ [22]. The publications reporting current studies of ecological succession were dated from 2007 to 2017, but the publications cited by the retrieved documents could be from any year. The whole network included a total of 29,398 publications of different types (articles, books, chapters, proceedings papers, etc.), with 245,210 citation connections among publications.

We excluded highly cited publications that described statistical tools or approaches because they did not describe any properties of the phenomenon of interest or the natural world. The most cited sources in this category were R Development Core Team [23] (110 citations, 791 summing the citations of different editions), Sneath et al. [24] (285 citations, 566 for all editions), McCune and Grace [25] (250 citations), Burnham and Anderson [26] (230 citations), Zar et al. [27] (197 citations).

To circumscribe a smaller set of the most referred publications in the studies of the collective of scientists (the 5,536 works identified in the previous step) we adopted a cut-off criterion based on the citation index and representativeness of publication. We added publications to the set of relevant publications sequentially, beginning with the most cited paper and then adding the following most cited, while checking the percentage of studies by the focal collective that cited this new set. The proportion of the 5,536 studies that cited at least one publication in the set tended to stabilize at 80% when the 25th most cited publication was included in the set.

We repeated this procedure considering only direct citations, considering direct and 2nd-degree citations (citations of citing articles), and considering direct, 2nd and 3rd degree citations. In all cases, there seemed to be a threshold around stabilization with 25 publications in the set of relevant publications (Fig 2). This analysis showed that about 20% of the studies that reflect the scientific activities of the focal collective were not citing the 100 most cited papers in the network.

We checked if this fraction (20%) of the studies could be hiding a subcommunity cohesively citing another set of relevant publications, not related to the ones most cited by the 80% of the collective. We created, then, a second network containing only those 20% publications that did not cite the 100 most cited in the first network, amounting to a total of 954 publications. Afterwards, we executed the same procedure to identify the most cited publications by these 20% publications. The most cited publications in this smaller network were cited by only 6% of the 954 publications network. This indicates that this fraction of 20% of studies sampled is not forming a divergent subcommunity with an alternative cohesive view about succession. Most likely, these publications form an heterogeneous group of papers that individually refer to lesser-known models in this field of study. Although we may technically consider these 20% as part of the collective of scientists studying succession, we cannot guarantee that their views about succession are contemplated by the 25 selected relevant publications, as they did not cite

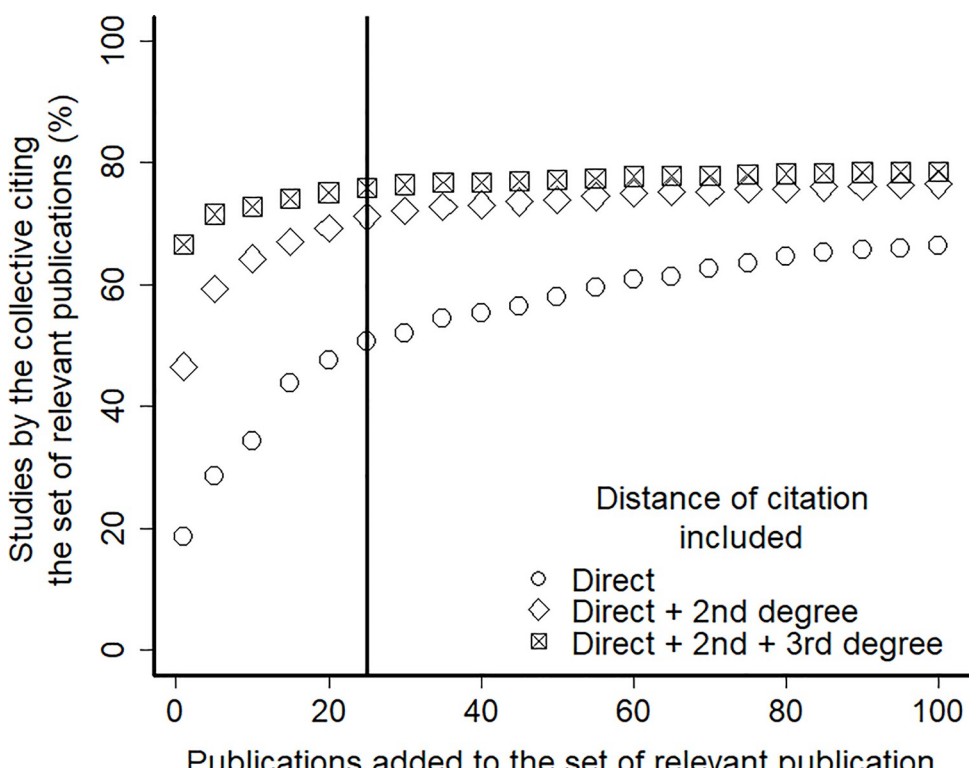

**Fig 2. Asymptotic relationship between the size of the relevant publication set and the percentage of studies authored by the defined collactive of scientists that cites the relevant publication set.** Direct citation distance considers only studies that cite the relevant publication directly. 2nd degree citations consider studies that directly cited the relevant publication and also studies that cited the studies that cited the relevant publications directly, the same logic applies to the 3rd degree citation. The vertical line indicates the threshold above which the percentage of studies of the collective that cite the set of relevant publications tends to stabilize, meaning that the addition of more relevant publications to the set does not aggregate more citations from the collective.

any of them. At the same time, because these studies do not refer to a well-defined alternative view of succession, it does not seem that including their views in the analysis will be useful to understand the major tendencies of the focal collective of scientists.

The results of this step indicate that a representative fraction of the publications reporting the scientific activities of the focal collective refers to at least one proposition made in the 25 most referred publications (Table 1). Thus, we adopted the 25 most referred publications as the ones containing the core conceptual bases for the studies on ecological succession by the defined community. The next step was to identify what propositions made in these publications are used by the collective of scientists as conceptual bases for studying succession.

## Excerpting

In this step, we selected a focal relevant publication and identified all studies of the collective of scientists that cited it. Then, we randomly selected one of these studies and searched for the excerpts of text in which the focal relevant publication was cited. We disregarded excerpts in which the relevant publication was cited along with other publications, to ensure that the excerpt of text was specifically referring to the content of the focal relevant publication. We stopped the inclusion of randomly selected studies once we reached 50 excerpts of text that fitted the criteria. Some of the sampled citing studies had more than one excerpt of text that fitted

**Table 1. List of relevant publications sorted by number of citations.**

| Publication | Direct citations |
| --- | --- |
| Connell and Slatyer 1977 [28] | 780 |
| Grime 1979 [29] | 468 |
| Connel 1978 [30] | 400 |
| Odum 1969 [31] | 374 |
| Harper 1977 [32] | 343 |
| Pickett and White 1985 [33] | 306 |
| Grubb 1977 [34] | 294 |
| Tilman 1988 [35] | 283 |
| Egler 1954 [36] | 271 |
| Cowles 1899 [37] | 266 |
| MacArthur and Wilson 1963 [38] | 264 |
| Bazzaz 1979 [39] | 207 |
| Huston 1979 [40] | 203 |
| Huston and Smith 1987 [41] | 200 |
| Grime 1977 [42] | 198 |
| Hubbell 2001 [43] | 189 |
| Pickett et al. 1987 [44] | 183 |
| Tilman 1987 [45] | 178 |
| Watt 1947 [46] | 178 |
| Huston and DeAngelis 1994 [47] | 177 |
| Grime 1988 [48] | 176 |
| Noble and Slatyer 1980 [49] | 171 |
| Tilman 1985 [50] | 163 |
| Shugart et al. 1984 [51] | 160 |
| Guariguata and Ostertag 2001 [52] | 160 |

The citations here are the ones made by the defined collective of scientists not the whole literature.

the criteria and others had none. We did this procedure to all 25 relevant publications, meaning that, at the end of this step, we had 50 excerpts of text for each of the 25 relevant publications. A pilot analysis revealed that 50 excerpts of text were enough to execute the next step (content analysis) efficiently.

## Analyzing the content of citations

The 50 excerpts of text citing each relevant publication from the previous step were then submitted to a content analysis, aiming at synthesizing the general propositions referred to in the relevant publications and used by the collective. Because the interpretation of a single person to create such a synthesis might introduce biases, we relied on an approach that considers intersubjectivity as key in the social processes of building scientific knowledge [53]. This analysis was carried out as follows:

1. **Setting a group of people to produce the syntheses based on the excerpts of text citing:** We recruited 25 volunteers, all of them graduate students or researchers in ecology. All the 25 volunteers have a degree in biology, eight had a PhD in ecology (two professors and six postdoctoral fellows of the department of ecology in the university where the study was carried out) and 17 were graduate students (six PhD candidates, and 11 master's degree candidates in ecology from the same university). The volunteers had different levels of familiarity

with the set of relevant publications, ranging from knowing, citing, discussing and teaching about the publication to no familiarity at all. Nevertheless, at the moment of the analysis, the volunteers were not informed about what publication the set of excerpts of text were referring to except that they belonged to a single relevant publication (the actual citations in the excerpts of text were replaced by a "[Relevant Publication]" tag, in order to inform where in the sentence the relevant publication was referred to).

2. **Dividing relevant publications among the volunteers:** Each set of 50 excerpts of text citing a single relevant publication was presented to three different volunteers, who formed a trio. Each trio was responsible for synthesizing the main propositions of three different relevant publications.

3. **Reading the excerpts of text:** For each relevant publication, each volunteer read the 50 excerpts. In reading the excerpts of text, the volunteers could find citations that did not establish a conceptual link between relevant publication and the citing study. The citations read might be just referring to a tool or method (in which case it was named 'operational'), it might not be used to structure an argument ('perfunctory'), or it might be used as an example of something wrong ('negational') (*sensu* Moravcsik & Murugesan [17]). The volunteers in our study were asked to disregard citations that were considered by them as operational, perfunctory or negational. Some of these three excluded categories of citation could help to identify the main contribution of a relevant publication. However, if a proposition is frequently cited in a domain, for example, in a negational way, it is unlikely that it is being used as a conceptual basis to create a new model to learn about the phenomenon of interest.

4. **Proposing individual syntheses:** Each volunteer produced a textual synthesis of what he/she thought was the main message of a focal relevant publication based only on their reading of the citing excerpts of text citing that relevant publication. This message could be a proposition, a set of propositions or a more complex mechanism composed of multiple propositions. Each volunteer was responsible for three relevant publications and, therefore, had to execute procedures 3 and 4 three times.

5. **Proposing group syntheses:** Each relevant publication received individual syntheses made by three different volunteers. The three syntheses were combined in a consensus activity adapted from the "Nominal Group Technique (NGT)" [54]. The goal of this activity was to synthesize a textual self consistent model or set of models that the three participants agreed upon as being what the citations of a focal relevant publication were referring to (see S1 File for methodological details of this step). For the 25 relevant publications, 29 verbal models about the natural world were identified.

As a result of this analysis, for each of the 25 relevant publications, one or more verbal model about ecological succession were identified. A verbal model is a set of propositions that together describe a phenomenon, mechanism, or process in nature. As such a verbal model can be composed of a single proposition, a list of independent propositions or a set of self-consistent complementary propositions. These sets were taken as the models from each relevant publication in use by the collective of scientists to learn about succession. The following text is an example of a model synthesized from the citations of the most cited relevant publication [28].

*The mechanisms of succession are interactions between individuals who colonize the environment first with those who colonize the environment after. These mechanisms are of three*

*types: (i) facilitation: in which species that colonize a place modify the environment, increasing the chances of colonization by other species; (ii) tolerance: where species that colonize a site do not affect the chances of establishing other species; (iii) inhibition: in which species that colonize a site modify the environment reducing the chances of colonization by other species. The relative importance of these mechanisms may vary over time due to changes in environmental conditions. The general functioning of these mechanisms is of a priority effect: the chances of colonization of a site by a species are affected by the species that colonized before this site. The model does not consider the routes and mechanisms of the arrival of the initial species (i.e., why that initial species is the initial species and not another).*

Through the PATh we were able to identify 29 verbal models about succession that were frequently and widely referred to in studies about ecological succession in the past decade. Athought the 29 verbal models are the main result of this study, each is at least one paragraph long and therefore describing everyone in the main manuscript would make it too long. All models in use that were recovered by the steps above are shown in S2 File.

## Pragmatic theory of succession

We present this case study using ecological succession to illustrate how the results obtained through the PATh can help us better understand ecology. In this section, we discuss three discoveries of the PATh that help make concepts more clear within ecological succession, tendencies of change within this field and divergences in this field brought to light by the PATh.

### Defining ecological succession

The models we identified make different statements about what ecological succession is and focus on different properties of the phenomenon. We are not interested, however, in particular views within the domain, but in defining the phenomenon of succession in a way that encompasses any of the models retrieved by PATh. To accomplish this task we analyzed a single description of succession coming from one of the central models and tried to identify the necessary and sufficient conditions for succession to occur, according to this description. Then, we analyzed a second model and checked if the conditions identified for the first description of succession were also necessary and sufficient for succession as described in the second central model. If not, we tried to describe the conditions in more general terms in a way they could also be considered necessary and sufficient for succession as described in the second central model. If this was not possible, we discarded that condition as being too specific for the general description of succession. We did this until all descriptions of succession found in the central models were analyzed. We reached two propositions about the necessary and sufficient conditions for any phenomenon of succession described in the set of central models to occur.

Proposition 1: At any moment in time, there is the possibility that resources will be available for use.

Proposition 2: Organisms from different species or at different ontogenetic stages have different probabilities of taking a fraction of the total available resource units. This difference can be due to (a) differential probabilities of site colonization, or (b) different probabilities that the individuals at the site or their propagules will take resource units.

However, some of the central models describe ecological processes that are different in many aspects. We then analyzed what was the primordial cause of the differences among these processes and conceived a third proposition that can explain why succession is conceived in such a diverse way.

Proposition 3: The dynamics of the resource and the probabilities of the species taking resource units are contingent on the abundance of species in the community and other environmental settings where the communities are changing.

Assuming that propositions 1 and 2 are necessary and sufficient for succession to occur means that in any situation in which these conditions are true, succession can occur, and if succession is occurring the two propositions hold. Even though the central models themselves may not have been conceived with these propositions explicitly in sight, our analyses show retrospectively that they underlie the construction of models within this field of study and, also, that by adopting them as true one could have conceived these models. Therefore, these propositions can be seen as fundamental principles of the domain. This means that any model that assumes these propositions as true is akin to the identified models and should be considered as a model within the domain of ecological succession as delimited by us. This information is not only useful to understand this field of study in which the PATh was applied. It helps ecologists decide if their phenomenon of study is succession or not, according to the focal collective considered in this study. This clarity can be used to avoid spurious debate, and can also reveal more straightforward ways to propose changes to how we conceive the phenomenon.

## Neutral models in succession theory

Five of the 29 identified models did not include the word "succession" in their description. Three of these models came from references about the competition-colonization trade-off and its relationship to disturbance regimes (Models 4, 16, and 22, described originally in [30], [40, 46], respectively. See S2 File). In the same vein, the two other models that did not mention succession explicitly were the models of Island Biogeography and the Neutral Theory, which describe the role of colonization and extinction in community assembly in general (Models 14 and 19, described in [38, 43]. See S2 File).

Even though these five models do not focus on describing or explaining the phenomenon of succession *per se*, they describe important concepts that are currently being used to make new propositions about what succession is and how it works. Considering that the domain of succession has been grounded in niche theory for many years [7], it is somewhat surprising that Hubbell's book [43] was the 16th most referred by the collective studying the phenomenon (being, in fact, more referred than some papers considered as classical references for studies on succession). An overview of the citing excerpts reveals that the neutral model is frequently used to explain successional patterns at landscape or global scales, meaning that this model is actually being used to learn about succession. Similarly, the model of island biogeography is frequently used to explain why different patterns of succession emerge in fragments at varying distances from other fragments (see S3 File). The fact that models assuming niche differentiation (e. g. [28, 29]) and models assuming competitive equivalence (e. g. [38, 43]) are being combined to learn about ecological succession, despite the incompatibility of such assumptions, strengthens our argument that theoretical syntheses based on axioms would not suffice to make an adequate description of this field of study [10]. An axiomatic synthesis of this field would have to either enunciate axioms that are not compatible with one another, something that would contradict the definition of axioms, or deliberately disregard some models that are in fact being used, neglecting how scientific research is actually being conducted. This could lead to discussions about the factuality of each contradicting axiom, as it has happened for these two kinds of models in the past, when we have evidence that both seem useful to learn about ecological phenomena [55–57].

## Classical and contemporary views on succession

We also observed that the division between "classical" and "contemporary" views on succession is not as clear for this collective of researchers as it is depicted in some publications, including textbooks (e. g. [19, 58–60]). Concepts classified as being from the classical view in these studies are still used today to develop new models and concepts classified as contemporary seems to be used in the present less frequently than one would expect. For example, in the "contemporary" view succession is regarded to be individualistic, while in the "classical" view succession is treated as supra-individualistic. Individualistic succession means that succession is essentially the exchange of individuals, in the sense that any pattern observed is just the result of interactions among these individuals. Supra-individualistic succession assumes that the agents of succession can be communities, functional groups or other supra-individual entities, and, therefore, succession can occur as these entities change in time [61]. The high frequency of citation of models considering supra-individual entities as the agents of succession (models 6 and 29, see S2 File. See [31, 52]) shows that such models of succession are used in the present, and not just rarely.

Odum's 1969 article was the 4[th] most cited publication in a network of more than five thousand items representing the scientific activity of the focal collective. This paper is frequently cited to support the idea that succession is the change in ecosystem properties through time. Hence, there can be little doubt that the scientific collective studying succession still adopts supra-individualistic approaches to this phenomenon as a possible way to understand it. Similarly, the model proposed in Guariguata and Ostertag [52] is used to give support to the claim that succession can be viewed as changes in functional groups, which are also supra-individual entities.

In the classical view, succession has been considered a dynamic intrinsic to communities of primary producers, while in the contemporary view, succession is considered a dynamic of communities at any trophic levels [19, 59]. Our findings show that the most refereed models in this field are still about the succession of plant communities. Most models are explicit about it and only a few propose mechanisms that could be applied to heterotrophic organisms. Therefore, it seems that succession ecology is still focused on plants and if there is a change towards a more multitrophic understanding of succession, this change is yet not noticeable enough to have surfaced with the most refereed models.

In synthesis, some models classified as belonging to the "classical view" are still being used often as a conceptual basis for learning more about succession in the last decade, while some others classified as belonging to the "contemporary view" have not been so frequently used. This result leads to the question of whether this classification of concepts as belonging to classical and contemporary views on succession is indeed a description of how scientists understand the theoretical development of this field or is rather some sort of rational reconstruction found in textbooks that do not correspond to the way scientists pragmatically use models, or, yet, a prescription of how ecological succession should be seen. These observations highlight some insights that the PATh allowed us to reach, in this case concerning the theoretical structure of the succession domain.

## Discussion

We showed that the PATh allows for a theoretical synthesis that assesses the conceptual bases of a field of study by describing the views of a defined community about an ecological phenomenon. Therefore, such a synthesis does fulfil one of the roles of a scientific theory, namely, the description of conceptual bases [62]. However, how the PATh allows for such a synthesis gives it some distinctive characteristics.

The first distinctive feature is that a synthesis made through the PATh has an explicitly descriptive character, in the sense that this synthesis intends to provide a summary description of how scientists are employing the knowledge available to them. This description arises, thus, from a pragmatic perspective on theories and models.

Pickett, Meiners & Cadenasso [19] provided a literature overview and a general theoretical synthesis for ecological succession that can be compared with the one produced by using the PATh. They also mention that their synthesis is to be used as a mechanistic reference in future studies of succession. They started from published papers to propose a synthesis around fundamental propositions about ecological succession and then described some central models in this field, just like we did. Some of their propositions agree with the ones derived from the PATh in some respects but disagree with others. It is important to note, however, that their synthesis was made by using an "expert opinion" approach which is highly dependent on the proponents' views and in which it is not clear what procedures were carried out to reach the set of propositions and models. These procedures need to be clear because without knowing them, it is not possible to be critical about the resulting synthesis and methodological criticism is one of the cornerstones in the construction of scientific knowledge [63].

The main difference of using the PATh instead of expert opinion to produce theoretical syntheses is that a detailed methodological description can be made to explain how propositions and models are directly linked to the actual scientific research made in a field. This process allows for a more direct and efficient process of criticism and re-evaluation of the synthesis. For example, one might argue that the execution of the third step in our case study was not conducted properly, resulting in a biased survey of the publications reflecting the activities of the focal collective of scientists. This could entail a set of relevant publications that do not contain conceptually important propositions about the phenomenon. This statement could then be tested, first, by evaluating if the parameters of the search may result in biased outcomes and then adjusting them to avoid the detected biases. Finally, we could check if the final set of central models obtained was in fact different from the one resulting from the previous search.

The second distinctive characteristic of syntheses made with the PATh is related to the assumption that a theory about a phenomenon can be seen as the views of a specific collective of scientists about a specific phenomenon or class of phenomena. This assumption implies that different collectives of scientists studying a phenomenon may have different theories about it. Because the definition of who is part of this collective is a methodological step in the PATh, multiple instances of the approach can be applied to different collectives and theoretical syntheses resulting from these different instances can be compared (or, perhaps, even combined to reach a more overarching approach). For example, an application of the PATh to different time scopes can reveal how a change may have happened through the disuse of models that were highly cited in the past and/or the increase in the use of models that are new. Scientific practices can also change from place to place [64, 65], as well as depending on gender [66], age, academic position and other social and cultural characteristics [67]. Different applications of the PATh could be used to answer if these different time, geographic and demographic profiles could lead to different views about a specific phenomenon. These social aspects of scientific work have been neglected for many decades in ecology [68] and in the natural sciences as a whole [69]. The PATh may offer, therefore, a tool for dealing with these aspects which may increase the interest in inspecting them more closely.

Finally, syntheses allowed by the PATh are explicitly descriptive in the sense that the enunciation of the central models identified is not a recommendation of the best models to learn about the focal phenomenon. These models merely summarize the conceptual basis of research conducted in this field of study. That, however, does not preclude syntheses produced

by the PATh to be used to guide scientific activity, although in a non-axiomatic way [10]. For example, we detected that many scientists in ecological succession accept that succession can be modelled as a supra-individualistic process, while others propose that succession be understood as an individualistic process [19]. This conflict revealed by the produced synthesis can alert either side to the need to express their viewpoints more specifically, either by presenting evidence supporting the claim that we should abandon models assuming supra-individualistic views of succession or arguing why there is no reason for that. If this knowledge is not yet available, it might be the case to invest in research to resolve this dispute. Such conduct should help avoid spurious debates within a field of study and, as a consequence, make this field more efficient in generating knowledge about the phenomenon. Because the PATh can be used to show which models are in fact being used, and how they are being used, if one thinks a model is being used for the wrong reasons, the PATh can help to spot and describe this use more precisely. Therefore, we can use the information obtained by accessing and analysing the views of a collective of scientists about a phenomenon using the PATh to make more informed decisions about productive changes and how to implement them. Changing is then assumed to be an integral part of theory development.

A synthesis made by using the PATh reveals the central models in use in a field of study and, therefore, it is an *a posteriori* synthesis. This is a shift from more axiomatized views of theories because it allows tinkering theories and models, merging or reinterpreting them, and so forth. Assuming that the conceptual basis of a field is grounded on a collective view of a group of scientists, any aspect of nature studied by scientists can be considered an object of the approach. It can be a general phenomenon, as succession is, but it also can be a specific mechanism operating within a phenomenon. Provided that one is able to identify the studies that are produced by scientific collective about a specific phenomenon (by using a systematic survey or otherwise), the approach will help to identify the most important models used to learn about that phenomenon. Imagine that someone is interested in producing a theoretical synthesis of, for instance, "conditions to stabilize mutualisms". It is perfectly feasible to define keywords to survey the studies about this specific mechanism. The definition of keywords can involve a little more debate than what we presented here because the words to define and delimit the studies about this mechanism might not be so easy to find (at least not as easy as "succession" was). In this case, one way to begin the approach would be to find some common concepts around the mechanism and then think of keywords to survey these concepts. After that, the approach can be carried out in the exact same way. The findings might be different from this case study, for example, resulting in models that do not lead to a set of unifying propositions or in a single model used to learn about the mechanism. Whichever the result is, the PATh will help make a decision on effective directions to proceed.

## Conclusion

Ecologists have shown that they can produce knowledge about the world without the restraints and guidance of a set of unifying axioms [70]. Some even disrecommended pursuing the systematization of knowledge into theories on the grounds that theories are reins that restrain scientific development [71]. Meanwhile, the number of studies proposing conceptual unification, disambiguation of concepts, conceptual cleaning and calls to training more theoreticians in ecology (e. g. [4, 72–76]) indicates that there is a demand for some kind of systematized way to represent the knowledge generated in ecology. Here we presented an approach that could help ecologists create theoretical syntheses without the necessity of identifying the set of unifying fundamental principles and fitting knowledge development into some predefined structure. These syntheses can be used to make more informed decisions about how to approach a

phenomenon, for example, by deciding to investigate more thoroughly a model that is gaining attention quickly, by investing in propositions that are being neglected by a community or by advising against the use of flawed models.

To the extent that the relevance of models within a field can be gauged by how much they are used by a collective of scientists, the relevance of a theoretical synthesis can also be gauged by how much a collective of scientists uses it. If ecologists do use the syntheses made by using the PATh to guide their scientific activities, these syntheses will fulfil both roles of a clearly enunciated theory, namely, describing and guiding knowledge generation [62]. Furthermore, since these syntheses are made *a posteriori* their guidance will work much more like consulting maps than restraining reins.

## Supporting information

**S1 File. Detailed steps of the Nominal Group Technique.**
(PDF)

**S2 File. Models in use identified by applying the PATh.**
(PDF)

**S3 File. Excerpts of text extracted from the studies of the collective of scientists citing the relevant publications.**
(PDF)

## Acknowledgments

We thank all 22 non-author participants that volunteered to the group-analysis (A. Palaoro, C. Hohlenwerger, D. Muniz, D. Bertuol, F. E. Mendes, F. D'Albertas, G. Bispo, G. Pitta, I. Romitelli, J. Menezes, L. Carneiro, L. Souza, L. Teixeira, M. Leite, N. H. Azevedo, R. Ourofino, R. Pelinson, R. Leporoni, R. Quesada, S. Iop, S. Mortara, V. Caldart). We thank D. Scarpa and M. E. Prestes for their comments on the initial version of this manuscript. We also thank the Academic Editors T. Heger and S. Consoli and the four anonymous reviewers for their comments and thoughtful handling of this manuscript in its final versions.

## Author Contributions

**Conceptualization:** Bruno Travassos-Britto, Renata Pardini, Charbel N. El-Hani, Paulo I. Prado.

**Data curation:** Bruno Travassos-Britto, Paulo I. Prado.

**Formal analysis:** Bruno Travassos-Britto, Renata Pardini, Charbel N. El-Hani, Paulo I. Prado.

**Funding acquisition:** Paulo I. Prado.

**Investigation:** Bruno Travassos-Britto, Renata Pardini, Charbel N. El-Hani, Paulo I. Prado.

**Methodology:** Bruno Travassos-Britto, Renata Pardini, Charbel N. El-Hani, Paulo I. Prado.

**Project administration:** Bruno Travassos-Britto.

**Supervision:** Paulo I. Prado.

**Validation:** Bruno Travassos-Britto.

**Visualization:** Bruno Travassos-Britto.

**Writing – original draft:** Bruno Travassos-Britto.

**Writing – review & editing:** Bruno Travassos-Britto, Renata Pardini, Charbel N. El-Hani, Paulo I. Prado.

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
