## [Decision Letter · Decision Letter 0]

9 Apr 2021

PONE-D-21-03811

A pragmatic approach to produce theoretical syntheses in ecology

PLOS ONE

Dear Dr. Travassos-Britto,

Thank you for submitting your manuscript to PLOS ONE. After careful consideration, we feel that it has merit but does not fully meet PLOS ONE’s publication criteria as it currently stands. Therefore, we invite you to submit a revised version of the manuscript that addresses the points raised during the review process.

We received two reviews for your manuscript. The reviewers come to quite opposing conclusions. In order to be able to come to a conclusion in a timely manner, I carefully read your manuscript myself in addition. Here is a summary of my own review:

As you stated in the submission letter, this manuscript is closely linked to another one submitted to the journal Oikos. While the latter frames your overall ideas concerning how ecology works as a science, and how theory is formed in this discipline, this manuscript introduces a method for identifying the conceptual basis for a specific area of interest. The method you propose is quite interesting in my point of view, and the case study applying it to the research field of succession produces relevant and interesting results.

My main concern is that readers may have difficulties in gaining access to your ideas. The introduction is quite philosophical, and I wonder whether this is even necessary. When reading the title of your paper, readers will very likely expect a proposal for a new synthesis method. And ecological synthesis usually does not start with axioms or theories, but with data and empirical results of single studies, trying to bring them together to see the big picture. I recommend you to frame the introduction more towards clarify how your method is situated in this context. Maybe the manuscript would even work well without any reference to the axiomatic approach?

Also, I was wondering if the method would be easier to understand if you would start with the case study. The chapter starting in line 72 is very abstract and I had problems figuring out how all this might work in practice. The brief summary of the relevant steps may work better in a chapter where you propose to go beyond the case study. There you could suggest to turn this into a method applicable to other cases as well. But this is just a suggestion - please try and see whether this works and is an improvement or not.

[As a side note, more related to the Oikos manuscript: your suggestions concerning a pragmatic, pluralist approach for ecology sound quite related to the following book, which you may not be aware of: Reiners, W. A., & Lockwood, J. A. (2010). Philosophical Foundations for the Practice of Ecology: Cambridge University Press.]

More comments are included in the attached pdf.

Now back to comments provided by the two reviewers. While Reviewer #1 regards your approach as very helpful and even points to its reproducibility as an achievement compared to other synthesis methods, Reviewer #2 considers your method as too subjective and prone for biases. In my opinion, conceptual synthesis is something that needs to be done by researchers using their creative potential and their language interpretation skills, and statistics can only be helpful to some degree here. I agree with Reviewer #1 that the consensus approach you chose is a helpful way to counteract subjectivity in this process. In order to avoid arguments like those put forward by Reviewer #2, you may try to clarify early on that the aim here is not to synthesize data or reach consensus about evidence, but that the kind of synthesis your method is designed for is conceptual synthesis. Your method thus (as far as I understood) does not aim at competing with formal, statistical meta-analyses, but rather with narrative conceptual reviews (which are often done with what you call the expert approach).

I very much agree with the comment by Reviewer #1 concerning generalizability of your approach. Please discuss this at some point in the manuscript. Ideally, you would come up with some suggestions for further applications of your method, to indicate whether it will always need to be such broad concepts, or whether you think this method could also work for synthesizing theory on more detailed research questions.

Also, please response to Reviewer #1’s comment concerning negative results and zombie ideas: Do you think these cases are covered during the step where you classified some of the studies as ‘negational’?

Please pay careful attention to all these aspects as well as additional suggestions made my the reviewers when revising your manuscript. I am looking forward to reading the revised version of your manuscript.

We look forward to receiving your revised manuscript.

Kind regards,

Tina Heger, PhD

Academic Editor

PLOS ONE

Journal Requirements:

Reviewers' comments:

Reviewer's Responses to Questions

**Comments to the Author**

1. Is the manuscript technically sound, and do the data support the conclusions?

Reviewer #1: Yes

Reviewer #2: Partly

2. Has the statistical analysis been performed appropriately and rigorously? 

Reviewer #1: N/A

Reviewer #2: N/A

3. Have the authors made all data underlying the findings in their manuscript fully available?

Reviewer #1: Yes

Reviewer #2: Yes

4. Is the manuscript presented in an intelligible fashion and written in standard English?

Reviewer #1: Yes

Reviewer #2: Yes

5. Review Comments to the Author

Reviewer #1: I think that the “Pragmatic approach to produce theoretical synthesis in ecology” is very useful and informative. As an ecologist, I found the approach logic and reproducible, which is an achievement given the level of subjectivity verbal summaries of theory are challenged.

I think the description of the approach is really straight forward. I really have no problem with the content of the paper. Most of my comments are on how some ideas are presented in the abstract and introduction. I had issues understanding the aim of the paper by reading only those parts. But these comments only require reformulating some language and maybe adding some extra explanations.

However, I do have a general question about the method. This approach is rather descriptive because is based on quantifying propositions “relevance” via number of citations (which is fully acknowledged by the authors), but how do you deal with “popular” but “outdated or wrong” propositions that have been perpetuated in current literature? For example, there was a discussion on ecology about the usefulness of the “intermediate disturbance hypothesis” stating that larger diversity is expected at intermediate levels of disturbance. The point of that discussion is that this hypothesis has been disproved several times, however, due to its simple and powerful prediction, it is still cited. Similar cases have been named “zombie ideas” that are still cited but they should no longer be. My point is, could this method run the risk of legitimizing propositions that are proven false on close examination, but people still cite and use them?

Another general question: is this method only useful for “defining” concepts. Your example is about how do you define “succession”. I think this is very useful, I can think of tons of concepts in ecology that could benefit from it (like “symbiosis”), however how could the method be used for more specific cases? For example, what about doing a theoretical synthesis on the “conditions that stabilize mutualisms”? or “the emergence of cheaters in symbiotic relationships in ecosystems”? Most commonly, reviews in ecology address such kind of questions, rather than a definition of a concept… do you think your method could also be extended to those cases? It would be great to have some lines on the generalizability of your approach in the discussion.

Abstract

I really think the abstract does not give merit to the paper. It is written in such a general way, that is unclear how unique is the approach. For example, I could not find the key argument that “proposition relevance” is measured by how “frequently those propositions are referred”.

I have problems with wording like “information and knowledge” or “what is being used by this collective”. It would be better to use “propositions” directly.

I think lines in the introduction (L47-50) and L60-66 would be great in the abstract. I think those lines summarize that a) ecology is pragmatic and not axiomatic, meaning that the most common propositions used by ecologist reflects a decision among these scientists on what verbal models are useful in order to understand a phenomenon; b) thus, a key challenge is to identify and quantify those propositions; c) your contribution is that you develop a method to achieve exactly that identification and quantification.

Introduction

L4: I suggest stating as sentence that the paragraph is about showing examples of how challenging is to make synthesis about theory in ecology. The reason for my suggestion is that after reading the sentence “However, the conceptual bases of knowledge generation in ecology might not be easy to identify” I was expecting an explanation of why is that the case. Instead, what is mentioned is a list of failed attempts to reach such identification.

L12. Related to my previous comment, I suggest here the first sentence should state that here is where the reason for this problem (the identification of the conceptual bases of ecology) is explained. Sth like: “Part of the problem in making theoretical synthesis in ecology is that scientists force this field to be axiomatic. We argue that this field is pragmatic.”

L70. Here you mention that you will discuss how your approach is different from “other approaches”. I thought that was really cool, but after reading the paper I realize you actually only compare to one specific case (the one used by Pickett, Meiners and Cadenasso). Unless I missed more approaches discussed, I think you should state within your case study, you compare your approach to another study that have similar aims but used a distinct approach.

L85. This is a typo; it should say “definition of the collective of agents”.

Reviewer #2: The manuscript by Travassos-Britto et al. discusses an important topic: how can we improve the way in which theoretical syntheses are made in ecology so as to meaningfully advance the field? The authors present an approach for producing theoretical syntheses based on the information most frequently used by researchers to learn about a phenomenon.

Their approach consists of six parts: 1) defining the phenomenon of interest; 2) defining a group of scientists studying the phenomenon; 3) performing a systematic literature search to identify the studies about the phenomenon published by this group; 4) identifying the most relevant publications cited in these studies; 5) identifying how the studies refer to the most relevant publications; 6) using a group of participants to synthesize what is being used by the group of scientists to learn about the phenomenon.

I appreciate the efforts of the authors for proposing a more methodological way to gather and summarise knowledge in ecology. I enjoyed reading about their example on succession and I found the findings reasonable based on my knowledge of the field.

My main concern is about the reproducibility of the findings. As the authors say, the synthesis produced will strongly depend on the collective of scientists that has been identified (step 2) as well as by the group of people used to synthesize the information (step 6). Unless multiple comparisons are made among different groups/regions/etc, the syntheses produced could be highly biased.

The lack of statistical or any kind of formal analyses at any step of the process increases this concern. I appreciate the idea of providing a more structured methodology to explain how knowledge and models are produced and used. The approach they propose could be used to allow a more direct and consistent comparison among reviews and syntheses of the field, but I am unsure of the generality of the syntheses produced as they would be strongly dependent on personal views since there is no objective or statistical way to analyse the data collected.

6. PLOS authors have the option to publish the peer review history of their article (what does this mean?). If published, this will include your full peer review and any attached files.

Reviewer #1: No

Reviewer #2: No

---

## [Author Response · Author response to Decision Letter 0]

15 Jul 2021

Dear Dr Heger, 

I’m submitting the revised version of the manuscript ID PONE-D-21-03811 entitled “A Pragmatic Approach to produce theoretical syntheses in ecology” to be considered for publication. My co-authors and I are very grateful for the insightful comments made by the reviewers and you. We genuinely appreciate how you have dealt with the revision process. The suggestions on how we should address some of the comments showed us that you read the manuscript carefully and genuinely want to improve the presentation of our idea. 

We agreed with most comments and also agree that the first version may not have been very clear for a readership of ecologists as it was heavy with philosophical jargon. To address the issues pointed out we made substantial changes in the introduction and abstract sections and changed the text of some other sections to better explain some of the methodological steps. 

In general, we think the reviewers and you identified some good points for debate and presented some legitimate questions about what we wanted to convey in this paper. We think that some criticisms related to the generality of the approach were caused by a misunderstanding of the aim of such an approach. We modified the text to make these points clearer. 

Please find also attached the file “Response to reviewers”, where we reply to each individual comment made by you and the reviewers and explain in detail the changes we made. This new version of the manuscript makes a much better presentation of our approach and we hope it is suitable for publication on PLOS One.

Thank you for considering this manuscript for publication and forwarding our reply to the reviewers.

Sincerely, 

Bruno Travasso-Britto 

bruno.travassos@usp.br

Editor:

Comment E1: 

The introduction is quite philosophical, and I wonder whether this is even necessary. When reading the title of your paper, readers will very likely expect a proposal for a new synthesis method. And ecological synthesis usually does not start with axioms or theories, but with data and empirical results of single studies, trying to bring them together to see the big picture. I recommend you to frame the introduction more towards clarifying how your method is situated in this context. Maybe the manuscript would even work well without any reference to the axiomatic approach?

Reply: We agree with that comment and substantially changed the introduction section. We removed the dichotomy between axiomatic and pragmatic views. In the current version, we focus mainly on why ecology could benefit from having a framework that is better aligned with its way of doing science. As a consequence, the introduction is much shorter and direct.

Comment E2:

Also, I was wondering if the method would be easier to understand if you would start with the case study. The chapter starting in line 72 is very abstract and I had problems figuring out how all this might work in practice. The brief summary of the relevant steps may work better in a chapter where you propose to go beyond the case study. There you could suggest to turn this into a method applicable to other cases as well. But this is just a suggestion - please try and see whether this works and is an improvement or not.

Reply: We understand that the description is somewhat abstract. We modified the text so the ideas in the description of the approach are less abstract. However, we decided not to change the order of the presentation of topics in the article because we worry that placing the case study first may convey the message that the specific method we used is what we are proposing as the approach. 

Our main message is that the proposed approach can be implemented in different ways, and thus the specific method we used in our study case is just one way to do that. Understanding this difference has other implications too.

We think that in a situation in which a reader thinks we are trying to propose a specific method, reservations about how we chose to conduct this method might lead her/him to discard the PATh approach as a whole. However, there is two main advantages in conveying the message that the case study is simply a proposal of how the PATh can be implemented. First, if a reader has reservations about the specific method we used, this perspective invites readers to think of better ways to implement the approach we proposed. Additionally, if readers propose better ways to implement the PATh, these proposals give strength to the argument that the approach indeed facilitates the revision process of synthesis generation. Therefore, we decided to maintain the structure of the manuscript because we believe this is the best way to present the general approach and the method used in the case study separately. 

Nevertheless we changed the text so that our intent to separate these things are clearer. 

Comment E3:

As a side note, more related to the Oikos manuscript: your suggestions concerning a pragmatic, pluralist approach for ecology sound quite related to the following book, which you may not be aware of: Reiners, W. A., & Lockwood, J. A. (2010). Philosophical Foundations for the Practice of Ecology: Cambridge University Press.

Reply: After you mentioned it, I (Bruno) read this book and am very glad I did. Although the discussions about pluralism are not as complete as in Nancy Cartright’s work, I found that it is completely aligned with Lorraine Daston works about intersubjectivity, and more focused on the ecological perspective too. We decided to cite the book to make some of the arguments in the text more specific to ecology.

It is somewhat surprising that we have this kind of discussion in the literature for decades, and yet, most manuscripts in which “theories” are being proposed are about a specific type of model, namely, mathematical model (as shown by Marquet et al 2014). 

Comment E4:

Reviewer #2 considers your method as too subjective and prone for biases. In my opinion, conceptual synthesis is something that needs to be done by researchers using their creative potential and their language interpretation skills, and statistics can only be helpful to some degree here. I agree with Reviewer #1 that the consensus approach you chose is a helpful way to counteract subjectivity in this process. In order to avoid arguments like those put forward by Reviewer #2, you may try to clarify early on that the aim here is not to synthesize data or reach consensus about evidence, but that the kind of synthesis your method is designed for is conceptual synthesis. Your method thus (as far as I understood) does not aim at competing with formal, statistical meta-analyses, but rather with narrative conceptual reviews (which are often done with what you call the expert approach).

Reply: Thank you for suggesting how to deal with the reviewer’s comment. As suggested, we added some sentences to make it clearer that the syntheses made by using the PATh are not meant to replace the approach that makes syntheses of data in section 2. However, we disagree that the lack of a statistical analysis is the same as the lack of a formal analysis and that a statistical analysis would make the synthesis less prone to personal biases. See the reply to Comment R2.1 and R2.2 in this document for more details of the argument.

Comment E5:

I very much agree with the comment by Reviewer #1 concerning generalizability of your approach. Please discuss this at some point in the manuscript. Ideally, you would come up with some suggestions for further applications of your method, to indicate whether it will always need to be such broad concepts, or whether you think this method could also work for synthesizing theory on more detailed research questions.

Reply: We agree that the generality of the approach needs to be presented more explicitly. We changed the last paragraph of the discussion to present the arguments we used to reply to Comment R1.2. Please, see the detailed response below. 

Comment E6:

Also, please response to Reviewer #1’s comment concerning negative results and zombie ideas: Do you think these cases are covered during the step where you classified some of the studies as ‘negational’?

Reply: Thank you again for the suggestion of how to address the issue. The classification of citations into negational or supportive was to guarantee that the propositions identified were, in fact, being used to support the scientific activities of the focal collective. The problem referred to in this comment will only be an issue if people do not understand the kind of data the PATh generates. It is important to understand that the PATh will identify what is being used, regardless of the reasons for that. This implies that sometimes, it will identify models that are being used for reasons that are not aligned with our views of how science should be. Therefore, the results of the PATh cannot be used to validate knowledge as empirically sound (see reply to Comment R1.1). However, there is a vast discussion in the epistemology of science about whether empirical support should be considered the most important epistemological value in scientific research. There are other values in science, such as heuristics, aesthetics, and simplicity. The PATh can be used to show that such models are in fact being used, and how they are being used. If one thinks a model is being used for the wrong reasons, we believe the PATh can help to spot and describe this more precisely, and thus will show what needs to be changed, according to that view.

In the current version of the manuscript, this aspect of this synthesis is discussed thoroughly in the 6th paragraph of the discussion section.

Reviewer 1

Comment R1.1:

This approach is rather descriptive because is based on quantifying propositions “relevance” via number of citations (which is fully acknowledged by the authors), but how do you deal with “popular” but “outdated or wrong” propositions that have been perpetuated in current literature? For example, there was a discussion on ecology about the usefulness of the “intermediate disturbance hypothesis” stating that larger diversity is expected at intermediate levels of disturbance. The point of that discussion is that this hypothesis has been disproved several times, however, due to its simple and powerful prediction, it is still cited. Similar cases have been named “zombie ideas” that are still cited but they should no longer be. My point is, could this method run the risk of legitimizing propositions that are proven false on close examination, but people still cite and use them? 

Reply: We completely understand this concern, and in fact we think our approach can contribute to spot propositions that are problematic, according to any view. Therefore, we do not think this will be a problem, provided people understand what the PATh is supposed to achieve. It is supposed to show us what models a determined community is using to learn about phenomena, independently of the reasons this community is using them. Because a community can make use of models for different reasons (which might be wrong reasons for a given view of science or theory), the information of which models are in use does not validate these models as empirically sound. However, this information can be used to propose changes and discuss future developments in the field.

For instance, the reviewer argued that the “intermediate disturbance hypothesis” should not be used to build new models because empirical evidence does not support it. Nevertheless, the PATh has shown that this model is frequently used, which may help to make the point on “zombie ideas”, if you wish. We think that the method can be useful to highlight the role of models for a given community, and thus allow us to scrutinize causes and consequences of such roles. We agree that the idea of “zombie ideas” is one among many possibilities, and we are glad to know that the reviewer had proposed this link. .

Comment R1.2:

is this method only useful for “defining” concepts. Your example is about how do you define “succession”. I think this is very useful, I can think of tons of concepts in ecology that could benefit from it (like “symbiosis”), however how could the method be used for more specific cases? For example, what about doing a theoretical synthesis on the “conditions that stabilize mutualisms”? or “the emergence of cheaters in symbiotic relationships in ecosystems”? Most commonly, reviews in ecology address such kind of questions, rather than a definition of a concept… do you think your method could also be extended to those cases? It would be great to have some lines on the generalizability of your approach in the discussion. 

Reply : We are very grateful for the reviewer for thinking of new ways in which our approach could be useful. We believe these examples could be addressed by using the PATh. We changed the text to make this possibility more clear in the text and also included the lines the reviewer suggested and, in fact, used one of the examples proposed by the reviewer.

 The main objective of the approach is to identify what models are being actually used to give support to studies conducted by a specified community of scientists. It is perfectly feasible to define keywords to survey the studies about “conditions to stabilize mutualisms”. After that, the steps of the PATh could be followed in the exact same way we conducted in our study to reach a set of models in use in this field. In the present study, we found it interesting to discuss the definition of the phenomenon by the focal collective, the dichotomy between what is in textbooks and what is being actually used, and how the neutral model is permeating a field that is historically based on niche. However, it is not warranted that the set of models used as conceptual basis in other fields of study will lead to the same observation we made. 

We’ve already applied the PATh to make a synthesis of the “species coexistence” domain. The definition of keywords involved a little more debate because there is no single word to define the phenomenon (at least not an obvious one like “succession”). We had to identify some common concepts around this phenomenon and then think of keywords to survey the studies about the phenomenon. After that, we followed the approach in the exact same way, which led us to a set of models in use. This set of models in use, as expected, is revealing facts about the theory in use about species coexistence that are different from what the models of the current manuscript revealed about the succession domain. We are currently writing the manuscript about this implementation of the PATh

Comment R1.3: 

I really think the abstract does not give merit to the paper. It is written in such a general way, that is unclear how unique is the approach. For example, I could not find the key argument that “proposition relevance” is measured by how “frequently those propositions are referred”.

Reply: We are thankful to the reviewer for making us aware of that. We changed the abstract so it is more descriptive of the Approach. This includes the information that the frequently referred publications are the ones used to survey the conceptual bases of the field of study.

Comment R1.4: 

I have problems with wording like “information and knowledge” or “what is being used by this collective”. It would be better to use “propositions” directly.

Reply: Agreed. In the previous version, we decided not to use simply “propositions” in some places because propositions are a statement that can be true or false, which is one of the ways in which knowledge transits among studies. However, we are essentially identifying propositions. Therefore, we believe that changing the manuscript as suggested greatly improved textual clarity without much loss of conceptual clarity. 

Comment R1.5: 

I think lines in the introduction (L47-50) and L60-66 would be great in the abstract. I think those lines summarize that a) ecology is pragmatic and not axiomatic, meaning that the most common propositions used by ecologist reflects a decision among these scientists on what verbal models are useful in order to understand a phenomenon; b) thus, a key challenge is to identify and quantify those propositions; c) your contribution is that you develop a method to achieve exactly that identification and quantification.

Reply: We appreciate this suggestion. The new version of the abstract tries to convey that information more clearly.

Comment R1.6:

L4: I suggest stating as sentence that the paragraph is about showing examples of how challenging is to make synthesis about theory in ecology. The reason for my suggestion is that after reading the sentence “However, the conceptual bases of knowledge generation in ecology might not be easy to identify” I was expecting an explanation of why is that the case. Instead, what is mentioned is a list of failed attempts to reach such identification.

L12. Related to my previous comment, I suggest here the first sentence should state that here is where the reason for this problem (the identification of the conceptual bases of ecology) is explained. Sth like: “Part of the problem in making theoretical synthesis in ecology is that scientists force this field to be axiomatic. We argue that this field is pragmatic.”

Reply: Agreed. Following the suggestion of the editor, we changed the text to make it more accessible to our readership. We removed the paragraph explaining in details the differences between axiomatic and pragmatic views, giving emphasis to the task of synthesizing ecological theory and how it can be made. We believe that this version has more internal coherence within paragraphs.

Comment R1.7:

L70. Here you mention that you will discuss how your approach is different from “other approaches”. I thought that was really cool, but after reading the paper I realize you actually only compare to one specific case (the one used by Pickett, Meiners and Cadenasso). Unless I missed more approaches discussed, I think you should state within your case study, you compare your approach to another study that have similar aims but used a distinct approach.

Reply: That is exactly the message we would like to convey, and we thank the reviewer for letting us know that it was not clear enough.. We changed the text to “ These results are used to discuss in what way our proposed approach to producing a theoretical synthesis is different from another study with the same aim but that used the more traditional “expert opinion” approach. We end up presenting how syntheses produced by using our approach can fulfil the role of a clearly enunciated theory.”

Comment R1.8

L85. This is a typo; it should say “definition of the collective of agents”.

Reply: This typo was corrected. 

Reviewer 2

Comment R2.1: I appreciate the efforts of the authors for proposing a more methodological way to gather and summarise knowledge in ecology. I enjoyed reading about their example on succession and I found the findings reasonable based on my knowledge of the field.

My main concern is about the reproducibility of the findings. As the authors say, the synthesis produced will strongly depend on the collective of scientists that has been identified (step 2) as well as by the group of people used to synthesize the information (step 6). Unless multiple comparisons are made among different groups/regions/etc, the syntheses produced could be highly biased.

Reply : We understand this concern as it was one of ours through the development of the approach and we agree that the previous version of the manuscript could be confusing in the methodological descriptions of the steps that dealt with these issues. We thank the reviewer for calling attention to these points. We significantly changed the section describing the content analysis of citations to convey more clearly why and how we conducted each of these steps. 

The dependence of the results of the approach on the collective of scientists and on the group of people used to synthesize the information are of different natures. The pragmatic view of theories assumes that scientists decide what to use to learn about phenomena, therefore, a different set of scientists might use a different set of models to learn about a phenomenon. This implies that different collectives of scientists may have different theories about a phenomenon. This is not a bias of the approach, this is a property of how theories develop according to the pragmatic view. 

 Multiple comparisons among different groups would indeed be very informative about particularities of how a determined phenomenon is studied across social groups. However, these multiple comparisons are not necessary to understand the views of the defined collective of scientists. The aim of the PATh is to identify what a determined community of scientists uses as conceptual bases to learn a phenomenon. Defining what community one intends to analyse is not assuming a bias it is defining the object of the study. 

 The group analysis step (6) is intended to control for known individual biases. Every theoretical synthesis has biases , and our approach is one of the few approaches to produce theoretical syntheses that acknowledges that and has a methodological step to make it less biased. 

The confidence in the final models is grounded on how we process actual data using an intersubjective approach (see Daston 1992 cited in the manuscript for a discussion about subjectivity in science). The citations are actual data coming from written documents that can be inspected. A group of people reads the same citations and makes individual syntheses based on this reading. The individual syntheses are as biased as any interpretation of text found throughout the literature. These individual syntheses are then subjected to a consensus analysis where they are compared for similarities and differences. After scrutinizing what was individual interpretation and real content coming from the citations, new collective syntheses are proposed. This reduces the individual biases of interpretation of the citations (see Bernard & Ryan, 2009, for more details on the assumptions of the method). 

Comment R2.2: The lack of statistical or any kind of formal analyses at any step of the process increases this concern. 

Reply: We did not conduct any quantitative analysis, but we did conduct a series of qualitative analysis to ground our results on actual data. It is important to note that our synthesis is not aimed at replacing data syntheses, such as meta-analysis. Rather, it is supposed to be complementary to such syntheses. However, it is important to remember that statistical analyses cannot give objectivity to a study. Statistical analysis uses a series of conventions to reach a consensus about quantities necessary to interpret data (see Daston 1992). We used a formal analysis to reach a qualitative consensus about data, this analysis allows us to have confidence in our interpretation of the data used, because it is standardized and reproducible, and also addresses sources of subjectivity.

 We also added some sentences regarding that matter in the section describing the general approach.

Comment R2.3: The approach they propose could be used to allow a more direct and consistent comparison among reviews and syntheses of the field, but I am unsure of the generality of the syntheses produced as they would be strongly dependent on personal views since there is no objective or statistical way to analyse the data collected.

Reply: We completely understand this concern, but we disagree that our approach is strongly dependent on personal views, for the reasons we detailed in the previous comment. As with any other approach to make conceptual syntheses, it still counts with human expertise and intuition. However, different from other approaches, ours is not only based on that. Another advantage of it is that the confidence in the result is not based on the renown of the ones proposing it, but on the method in itself. There are different ways to execute each step, therefore, if one is not confident in the results, she/he can point out which step of the method has a problem.

---

## [Decision Letter · Decision Letter 1]

19 Aug 2021

PONE-D-21-03811R1

A pragmatic approach to produce theoretical syntheses in ecology

PLOS ONE

Dear Dr. Travassos-Britto,

Thank you for submitting your manuscript to PLOS ONE. After careful consideration, we feel that it has merit but does not fully meet PLOS ONE’s publication criteria as it currently stands. Therefore, we invite you to submit a revised version of the manuscript that addresses the points raised during the review process.

One of the previous reviewers agreed to take a look at the revision, and is quite satisfied with your revision and responses. I agree, and thank you for this thorough revision, addressing all the raised issues. The reviewer made a few more comments, which I would like to ask you to consider before we can proceed to the final stages. Also, I found a tiny mistake: in line 500, something is missing at the end of the sentence.

I am looking forward very much to seeing this important paper published.

We look forward to receiving your revised manuscript.

Kind regards,

Tina Heger, PhD

Academic Editor

PLOS ONE

Journal Requirements:

Reviewers' comments:

Reviewer's Responses to Questions

**Comments to the Author**

1. If the authors have adequately addressed your comments raised in a previous round of review and you feel that this manuscript is now acceptable for publication, you may indicate that here to bypass the “Comments to the Author” section, enter your conflict of interest statement in the “Confidential to Editor” section, and submit your "Accept" recommendation.

Reviewer #1: (No Response)

2. Is the manuscript technically sound, and do the data support the conclusions?

Reviewer #1: Yes

3. Has the statistical analysis been performed appropriately and rigorously? 

Reviewer #1: N/A

4. Have the authors made all data underlying the findings in their manuscript fully available?

Reviewer #1: Yes

5. Is the manuscript presented in an intelligible fashion and written in standard English?

Reviewer #1: Yes

6. Review Comments to the Author

Reviewer #1: I think the authors have done a great job improving the manuscript entitled: “A pragmatic approach to produce theoretical syntheses in ecology”. All my previous comments have been addressed by the authors (either in their reply to my comments or in the revised manuscript). At this point I only have two minor suggestions:

1. Regarding my comment about how PATh can help pointing out inconsistencies/wrong retionale in the models used around a phenomenon (my comment on “zombie” ideas):

I think the authors should add in the manuscript this sentence from the response to my comment:

“The PATh can be used to show that such models are in fact being used, and how they are being used. If one thinks a model is being used for the wrong reasons, we believe the PATh can help to spot and describe this more precisely, and thus will show what needs to be changed, according to that view.”

I like what you already wrote regarding this point in the discussion (L529-546), however that sentence you wrote in the reply states much better, in my opinion, how PATh can address the wrong use of models.

2. Regarding my comment on how to use PATh beyond definition of Concepts:

I think what you wrote in the response is great, and it should be added. Specfically the part about using keywords which basically adds a new step. I think this is very interesting information and, thus I suggest adding to the discussion these lines (after adjusting them of course):

“It is perfectly feasible to define keywords to survey the studies about “conditions to stabilize mutualisms” […] The definition of keywords involved a little more debate because there is no single word to define the phenomenon (at least not an obvious one like “succession”). We had to identify some common concepts around this phenomenon and then think of keywords to survey the studies about the phenomenon. After that, we followed the approach in the exact same way, which led us to a set of models in use.”

7. PLOS authors have the option to publish the peer review history of their article (what does this mean?). If published, this will include your full peer review and any attached files.

Reviewer #1: No

---

## [Author Response · Author response to Decision Letter 1]

23 Aug 2021

Reviewer #1: I think the authors have done a great job improving the manuscript entitled: “A pragmatic approach to produce theoretical syntheses in ecology”. All my previous comments have been addressed by the authors (either in their reply to my comments or in the revised manuscript). At this point I only have two minor suggestions:

1. Regarding my comment about how PATh can help pointing out inconsistencies/wrong retionale in the models used around a phenomenon (my comment on “zombie” ideas):

I think the authors should add in the manuscript this sentence from the response to my comment:

“The PATh can be used to show that such models are in fact being used, and how they are being used. If one thinks a model is being used for the wrong reasons, we believe the PATh can help to spot and describe this more precisely, and thus will show what needs to be changed, according to that view.”

I like what you already wrote regarding this point in the discussion (L529-546), however that sentence you wrote in the reply states much better, in my opinion, how PATh can address the wrong use of models.

2. Regarding my comment on how to use PATh beyond definition of Concepts:

I think what you wrote in the response is great, and it should be added. Specfically the part about using keywords which basically adds a new step. I think this is very interesting information and, thus I suggest adding to the discussion these lines (after adjusting them of course):

“It is perfectly feasible to define keywords to survey the studies about “conditions to stabilize mutualisms” […] The definition of keywords involved a little more debate because there is no single word to define the phenomenon (at least not an obvious one like “succession”). We had to identify some common concepts around this phenomenon and then think of keywords to survey the studies about the phenomenon. After that, we followed the approach in the exact same way, which led us to a set of models in use.”

Reply: We agree that some of the statements we made in the response letter are in some ways more straightfoward and clear than what we used in the text. As suggested, we added the indicated sentences to the main manuscript text (see lines 542-545 and lines 558 - 565). We agree that, now, these paragraphs give a much more concrete explanation.

---

## [Decision Letter · Decision Letter 2]

28 Sep 2021

PONE-D-21-03811R2A pragmatic approach to produce theoretical syntheses in ecologyPLOS ONE

Dear Dr. Travassos-Britto,

Thank you for submitting your manuscript to PLOS ONE. After careful consideration, we feel that it has merit but does not fully meet PLOS ONE’s publication criteria as it currently stands. Therefore, we invite you to submit a revised version of the manuscript that addresses the points raised during the review process.

The paper has improved evidently and the contents are worth of interest for the community. There are however still points to be addressed before the manuscript can reach an acceptable standard for being published. 

The material in the paper looks interesting, but the manuscript need to be further revised carefully to meet PLOS ONE publication criteria. 

Please take particular care to the comments raised by Reviewer 4.

In particular make sure to have data available, you should present the "real data" used on your synthesis.

In terms of the results of the synthesis, the paper could benefit from more context and explanation, proving that your final results give evidence that your synthesis was the best approach in your analysis.  

Please take carefully into account the comments of all the referees for improving the manuscript to meet PLOS ONE standards before resubmitting it to the journal.

We look forward to receiving your revised manuscript.

Kind regards,

Sergio Consoli

Academic Editor

PLOS ONE

Reviewers' comments:

Reviewer's Responses to Questions

**Comments to the Author**

1. If the authors have adequately addressed your comments raised in a previous round of review and you feel that this manuscript is now acceptable for publication, you may indicate that here to bypass the “Comments to the Author” section, enter your conflict of interest statement in the “Confidential to Editor” section, and submit your "Accept" recommendation.

Reviewer #1: All comments have been addressed

Reviewer #3: (No Response)

Reviewer #4: (No Response)

2. Is the manuscript technically sound, and do the data support the conclusions?

Reviewer #1: Yes

Reviewer #3: Yes

Reviewer #4: Partly

3. Has the statistical analysis been performed appropriately and rigorously? 

Reviewer #1: N/A

Reviewer #3: N/A

Reviewer #4: I Don't Know

4. Have the authors made all data underlying the findings in their manuscript fully available?

Reviewer #1: (No Response)

Reviewer #3: Yes

Reviewer #4: No

5. Is the manuscript presented in an intelligible fashion and written in standard English?

Reviewer #1: Yes

Reviewer #3: Yes

Reviewer #4: Yes

6. Review Comments to the Author

Reviewer #1: (No Response)

Reviewer #3: I have enjoyed reading this MS despite holding the belief that such frameworks of theories may indeed, as the authors point out, sometimes restrain knowledge and development of a field such as ecology.

It is however clear that the authors have endeavoured to satisfy previous reviewer comments, notably on the clarity of the manuscript, which is now appropriate for publication in my opinion. I do not find the current MS too philosophical, the language used is apt, and should be understood and appreciated by a wide audience. Moreover, the figures well represent the MS and add clarity. I would therefore recommend this MS for publication, though I do have a few suggestions that ought to be addressed first.

I agree with previous reviewers that the abstract still does not add merit to the MS. In particular, the two final sentences are too vague. It is important to state here what PATh enables/improves specifically that was previously not possible/clear.

I agree with the editors comment that the method is somewhat prone to bias, please add a sentence to the discussion to clarify if the outcome might have differed using a different makeup (age, level of qualification/position, interest) of researchers.

Some small changes to be made at the authors discretion as follows:

1. Consider changing ‘produce’ to ‘producing’ in the title.

2. Line 52 and elsewhere ‘the field’ to most ecologists means surveying/sampling. Please specify ‘this field’ or ‘ecology’

3. Line 53 consider changing ‘scientific activity in the area’ to ecological activity/science’

4. Line 78 make clear that this general approach can be applied to specific topics/sub-fields within ecology.

5. Line 175 ‘path’ should be ‘PATh’.

6. Line 226 ‘created a cut-off’ should be ‘used/adopted a cut-off’

7. Line 275 ‘Thus, we adopted’

8. Line 580 ‘Hacking’ reference should be reformatted.

9. Throughout the article, ‘Fig’ should be ‘Fig.’

10. Supplementary 1, Point 3. ‘Each proposition’

Reviewer #4: I was excited by the idea of this paper. I feel that this could be a unifying way to synthesize the literature in ecology. However, I felt that the paper had several shortcomings:

1)Need to better set up the theory in ecological succession before using it as the example. Why ecological succession chosen for this study? Need more background here.

2) Need to provide the results. Is there a table of the results of your analyses. How many paper fell into each category by evaluators? How were the citations distributed? What were the most common propositions of relevant publications? What were common themes in textual syntheses? What were the 29 textual models?

3) You in text example could be taken out of paper, Lines 341-353. I did not follow what you synthesized from the text and how you identified the 29 verbal models? I don't follow some of your terminology. Verbal models? textual models? how do they differ? These need to be better described.

4) Your "Propositions" (Lines 380-392): I found these to be confusing. Why are they not written in terms of the ecological theory in ecology. The way that they are phrased does not sound like ecology.

5) What is your final synthesis? What models are best? I need the "so what" for ecology. State the important gains from your study.

Any meta-analyses?

7. PLOS authors have the option to publish the peer review history of their article (what does this mean?). If published, this will include your full peer review and any attached files.

Reviewer #1: No

Reviewer #3: No

Reviewer #4: No

---

## [Author Response · Author response to Decision Letter 2]

28 Oct 2021

Editor:

Comment E2.1: 

The paper has improved evidently and the contents are worth of interest for the community. There are however still points to be addressed before the manuscript can reach an acceptable standard for being published. 

The material in the paper looks interesting, but the manuscript need to be further revised carefully to meet PLOS ONE publication criteria. 

Please take particular care to the comments raised by Reviewer 4.

In particular make sure to have data available, you should present the "real data" used on your synthesis.

Reply: All data used in the analysis is described in the manuscript or in the supplementary material. We believe that the perception that the whole data was not available was caused by a mislabelling of supplementary material, which has been corrected. Please see a more thorough explanation in the replies to Comment R4.2. 

Comment E2.2

In terms of the results of the synthesis, the paper could benefit from more context and explanation, proving that your final results give evidence that your synthesis was the best approach in your analysis. 

Reply: We understand that most manuscripts describing methodological approaches are aimed at proving that the described approach is in fact the best when compared to other approaches. However, what makes this approach more adequate to deal with the theoretical synthesis in ecology is the fact that it is based on the pragmatic view of theories. As we described in the introduction, it is difficult to produce a theoretical synthesis of a field that develops pragmatically. Nevertheless, ecology develops pragmatically as we argue in Travassos-Britto et al, 2021. Our approach is the only one, to our knowledge, that is aligned with the pragmatic view of theories in ecology. Therefore, the objective of the manuscript is to argue that it is possible to implement such an approach, to give examples of the kind of information we can generate with it, and to describe how different this approach is from other approaches because of the pragmatic assumption. In other words, our results from the case study are not intended to prove that this is the best approach, they are intended to exemplify how we can use the approach to further knowledge in this field.

 The structure of the manuscript was thought to convey this message. In the introduction, we set the problem with syntheses based on axiomatic views of theories, in the “PATh – Pragmatic Approach to Theories” we describe the general premises behind the approach; In the “A case study on ecological succession” we describe the way we choose to implement it; in the “Pragmatic theory of succession” we describe the types of insights we can have by using the method and in the “Discussion” section we present how different this approach is from approaches based on the axiomatic premises. 

We modified the new version of the manuscript to make it clear. In the new version, the beginning of each of these sections makes it clear that this is the objective of this manuscript. 

Reviewer 3

Comment R3.1:

I have enjoyed reading this MS despite holding the belief that such frameworks of theories may indeed, as the authors point out, sometimes restrain knowledge and development of a field such as ecology.

Reply: We understand the view that theories restrain knowledge development because theory is used as a guide and this may make it more difficult to see what is not described in the guide. However, we believe that syntheses produced by this approach are definitely less restraining than syntheses proposed by other known approaches. First, because PATh syntheses are explicitly descriptive. In this sense, they state that this is what people are doing, not necessarily how the user of the synthesis should do. Second, the synthesis is based on the premise that we don’t need theories unified by axioms to produce knowledge. Therefore, in a way, this kind of synthesis encourages scientists to make decisions based on what they think it’s useful to learn about the natural world. Third, if theories may restrain knowledge development in some sense, we also cannot truly progress in producing knowledge without the conceptual and methodological focus provided by theories. Thus, it is not so much about regretting that theories guide our views on phenomena, but about finding the proper balance between the focused view theories allow us and the shifts that transformations of our perspectives on phenomena can give us.

Comment R3.2:

It is however clear that the authors have endeavoured to satisfy previous reviewer comments, notably on the clarity of the manuscript, which is now appropriate for publication in my opinion. I do not find the current MS too philosophical, the language used is apt, and should be understood and appreciated by a wide audience. Moreover, the figures well represent the MS and add clarity. I would therefore recommend this MS for publication, though I do have a few suggestions that ought to be addressed first.

I agree with previous reviewers that the abstract still does not add merit to the MS. In particular, the two final sentences are too vague. It is important to state here what PATh enables/improves specifically that was previously not possible/clear.

Reply: We appreciate that the reviewer acknowledged our effort towards reaching a wider audience, as we detailed in our comments to the Editor. We made modifications in the abstract to be more precise in what differentiates this approach from other ways to produce syntheses.

Comment R3.3:

 I agree with the editors comment that the method is somewhat prone to bias, please add a sentence to the discussion to clarify if the outcome might have differed using a different makeup (age, level of qualification/position, interest) of researchers.

Reply: We are not sure if the reviewer refers to a bias caused by the makeup of the community of scientists under study or whether it refers to a bias caused by the makeup of the volunteers involved in producing the syntheses. If it’s the latter, we agree that the outcome may vary with the makeup of the synthesis group. We think this is an expected feature of our approach, which is conceived to allow groups to build their own syntheses on how theory of a given field is (or was) structured. Of course, in many cases it is to share such syntheses to a wider audience, but as the approach provides all data (e.g. references, excerpts) the resulting synthesis is open to scrutiny. The item 1 of the nominal group technique description states about the risk of this bias (see lines 300-312 of the Main MS)

 If the bias is referring to the makeup of the scientific community under study. I think the reply to the Comment R2.1 of the previous round addresses this issue in a very thorough manner. I pasted it below for easier consultation.

[[Comment R2.1: I appreciate the efforts of the authors for proposing a more methodological way to gather and summarise knowledge in ecology. I enjoyed reading about their example on succession and I found the findings reasonable based on my knowledge of the field.

My main concern is about the reproducibility of the findings. As the authors say, the synthesis produced will strongly depend on the collective of scientists that has been identified (step 2) as well as by the group of people used to synthesize the information (step 6). Unless multiple comparisons are made among different groups/regions/etc, the syntheses produced could be highly biased.

Reply: We understand this concern as it was one of ours through the development of the approach and we agree that the previous version of the manuscript could be confusing in the methodological descriptions of the steps that dealt with these issues. We thank the reviewer for calling attention to these points. We significantly changed the section describing the content analysis of citations to convey more clearly why and how we conducted each of these steps. 

The dependence of the results of the approach on the collective of scientists and on the group of people used to synthesize the information are of different natures. The pragmatic view of theories assumes that scientists decide what to use to learn about phenomena, therefore, a different set of scientists might use a different set of models to learn about a phenomenon. This implies that different collectives of scientists may have different theories about a phenomenon. This is not a bias of the approach, this is a property of how theories develop according to the pragmatic view. 

 Multiple comparisons among different groups would indeed be very informative about particularities of how a determined phenomenon is studied across social groups. However, these multiple comparisons are not necessary to understand the views of the defined collective of scientists. The aim of the PATh is to identify what a determined community of scientists uses as conceptual bases to learn a phenomenon. Defining what community one intends to analyse is not assuming a bias it is defining the object of the study. 

 The group analysis step (6) is intended to control for known individual biases. Every theoretical synthesis has biases , and our approach is one of the few approaches to produce theoretical syntheses that acknowledges that and has a methodological step to make it less biased. 

The confidence in the final models is grounded on how we process actual data using an intersubjective approach (see Daston 1992 cited in the manuscript for a discussion about subjectivity in science). The citations are actual data coming from written documents that can be inspected. A group of people reads the same citations and makes individual syntheses based on this reading. The individual syntheses are as biased as any interpretation of text found throughout the literature. These individual syntheses are then subjected to a consensus analysis where they are compared for similarities and differences. After scrutinizing what was individual interpretation and real content coming from the citations, new collective syntheses are proposed. This reduces the individual biases of interpretation of the citations (see Bernard & Ryan, 2009, for more details on the assumptions of the method). ]]

Comment R3.4:

Some small changes to be made at the authors discretion as follows:

1. Consider changing ‘produce’ to ‘producing’ in the title.

2. Line 52 and elsewhere ‘the field’ to most ecologists means surveying/sampling. Please specify ‘this field’ or ‘ecology’

3. Line 53 consider changing ‘scientific activity in the area’ to ecological activity/science’

4. Line 78 make clear that this general approach can be applied to specific topics/sub-fields within ecology.

5. Line 175 ‘path’ should be ‘PATh’.

6. Line 226 ‘created a cut-off’ should be ‘used/adopted a cut-off’

7. Line 275 ‘Thus, we adopted’

8. Line 580 ‘Hacking’ reference should be reformatted.

9. Throughout the article, ‘Fig’ should be ‘Fig.’

10. Supplementary 1, Point 3. ‘Each proposition’

Reply: All suggestions were accepted. We are thankful to this reviewer for the careful reading of our manuscript.

Reviewer 4

I was excited by the idea of this paper. I feel that this could be a unifying way to synthesize the literature in ecology. However, I felt that the paper had several shortcomings:

Comment R4.1

1)Need to better set up the theory in ecological succession before using it as the example. Why ecological succession chosen for this study? Need more background here.

Reply: Agreed. In the section “Defining the phenomenon of interest”, in which we explain why we chose ecological succession as the object of study, we added an explanation regarding the domain history. In fact, one of the reasons why we chose this phenomenon was its long history and its numerous theoretical syntheses proposed in the past. We changed the manuscript to make such reasoning clearer. 

Comment R4.2

2) Need to provide the results. Is there a table of the results of your analyses. 

Reply: All data used in the study are available either in the manuscript or as supplementary material. We identified that the previous versions of the manuscript had some mislabelling of supplementary material. We apologize for this issue, which is now corrected. 

Regardless, we think it is worth clarifying that the manuscript describes a new approach thought to be aligned with the pragmatic view of theories and, therefore, it might not follow any previously used standard description of methods. We will take this opportunity to respond to each question individually explaining where we made results available and why.

Comment R4.2.1:How many paper fell into each category by evaluators? 

Reply: The papers were not divided into categories. Each relevant publication had a set of citing publications that could vary in size. The following description was conducted by each of the relevant publications. First, we identified the set of publications of the scientific community that cited the focal relevant publication. Then, in these publications, we identified phrases or paragraphs in which the focal relevant publication was cited and excerpted these snippets of text referring to the relevant publication. For each relevant publication, there were 50 of these snippets excerpted from the publications published by the focal community. To produce an individual synthesis of the model in use originated in a relevant publication, a volunteer read these 50 excerpts of text. All excerpts of text read by the volunteers are described in Supplementary Material 3 labelled by the relevant publication they refer to. This process is completely described in detail in the section “Analyzing the content of citations” in topics (lines 300-341 of the Main MS).

Comment R4.2.2:How were the citations distributed? 

Reply: Citations were distributed according to what is described in Table 1 of the MS (line 285). It is important to note that these are not general citations of each relevant publication. These are the citations made by the focal scientific community under study. We changed the caption of the table to make it more clear.

Comment R4.2.3: What were the most common propositions of relevant publications?

Reply: The most relevant propositions of each relevant publication are described in the textual models. A textual model can be composed of a single proposition or a set of propositions that work together as a nomological machine. We describe this concept in detail in the reply to Comment R4.3.2, we also added a more clear definition in the MS (342-348). 

Comment R.4.2.4: What were common themes in textual syntheses?

Reply: Although we agree that a thematic analysis could reveal interesting things about the similarities among textual models, we did not believe that such an analysis would be necessary to reach our goal. We used a consensus analysis based on an adaptation of a nominal group technique. The exact method we used is described in the Supplementary material 1. 

Comment R4.2.5: What were the 29 textual models?

Reply: Each of the 29 models is described in supplementary material 2. We agree that the main result of the approach is the set of 29 textual models and it might be unorthodox to list it in the supplementary material. However, each of the 29 models is at least one paragraph long. Therefore, adding it to the main text could make it too long for a standard paper of PLOS One. We included this information in the main manuscript (Lines 369-371), so it is clear to readers that the main results are in the Supplementary Material.

Comment R4.3

Comment R4.3.1: 3) You in text example could be taken out of paper, Lines 341-353. I did not follow what you synthesized from the text and how you identified the 29 verbal models? 

Reply: I believe some lack of clarity is justified in the fact that there was some mislabeling of the supplementary material in the MS. This is actually one example of the 29 models reached. This textual description of a mechanism was the result of following steps 1 to 6 of the approach. This text was synthesized by the volunteers based only on their reading of the excerpts of text referring to Connel & Slatyer (1977). It is important to note that the evaluators did not know to which publication the excerpts of text were referring to when they read it and some evaluators have had none or almost no contact with this literature. Therefore, this verbal model is the actual result of a synthesis based only on the reading of citing excerpts of texts referring to Connel & Slatyer (1977). It is not (and should not be) based on the previous knowledge of what Connel and Slatyer wrote in their paper. The process to generate such model is described in detail in the section “Analyzing the content of citations”

We used this model in the MS to exemplify what are the verbal models resulting from this process, the rest of the models are described in supplementary material 2. We think the clearer definition of the verbal models (lines 343-346) and the clearer explanation that the other models are in the Supplementary Material should make this clear.

Comment R4.3.2: I don't follow some of your terminology. Verbal models? textual models? how do they differ? These need to be better described.

Reply: We agree that there were no clear definitions for these terms. We were using them as synonyms. We changed the MS to use just “verbal models” and gave a proper definition in lines 336-340. Thank you for alerting us.

Comment R4.4

4) Your "Propositions" (Lines 380-392): I found these to be confusing. Why are they not written in terms of the ecological theory in ecology. The way that they are phrased does not sound like ecology.

Reply: We understand that these propositions are not written as one might find in classical literature in ecology. The phrasing is a result of the method described in the first paragraph of the section “Defining ecological succession”.We reassessed the method and the sentences to check if we could change it to a more familiar phrasing. We found the phrasing of these propositions the most direct and unequivocal way to make the statements without contradicting any of the 29 models identified. 

Comment R4.5

 What is your final synthesis? What models are best? I need the "so what" for ecology. State the important gains from your study.

Reply: We think that the reviewer was precluded from understanding all our results due to the mislabeling of the supplementary material. We corrected this issue and we think that these questions are properly addressed in the paper. 

In short, the objective of the approach is not to indicate the best models to learn about a phenomenon. It is rather to map how a specific community of scientists is using models to learn about the phenomenon. The resulting verbal models synthesize the views of the selected community about ecological succession. These models reveal major tendencies of model use within the scientific community, as we discussed in the section “Pragmatic theory of succession”. This, in no way, is a recommendation of the best models to understand succession (as described in the second to last paragraph of the Discussion section). However, understanding these tendencies gives us a clear picture of the state of research within a domain of study, which should help us better guide the future of the domain.

Comment R4.6: 

Any meta-analyses?

Reply: We do think that a meta-analysis could be an interesting way to contrast what we’ve found with quantitative data. The meta-analysis and the PATh, however, have very distinct objectives. While the former is an approach that assesses confidence in statistical analysis about a hypothesis, the latter is an approach to assess how models and their subjacent propositions are being used as conceptual bases to learn about a phenomenon. At the moment, we are studying how we can combine the two approaches to check if quantitative data gives support to what is being used by the scientific community. I think the reply to Comment R2.2 from previous rounds of review can give a clear insight into how we see it.

[[[Comment R2.2: The lack of statistical or any kind of formal analyses at any step of the process increases this concern. 

Reply: We did not conduct any quantitative analysis, but we did conduct a series of qualitative analysis to ground our results on actual data. It is important to note that our synthesis is not aimed at replacing data syntheses, such as meta-analysis. Rather, it is supposed to be complementary to such syntheses. However, it is important to remember that statistical analyses cannot give objectivity to a study. Statistical analysis uses a series of conventions to reach a consensus about quantities necessary to interpret data (see Daston 1992). We used a formal analysis to reach a qualitative consensus about data, this analysis allows us to have confidence in our interpretation of the data used, because it is standardized and reproducible, and also addresses sources of subjectivity.

 We also added some sentences regarding that matter in the section describing the general approach.]]]

---

## [Decision Letter · Decision Letter 3]

29 Nov 2021

A pragmatic approach for producing theoretical syntheses in ecology

PONE-D-21-03811R3

Dear Dr. Travassos-Britto,

We’re pleased to inform you that your manuscript has been judged scientifically suitable for publication and will be formally accepted for publication once it meets all outstanding technical requirements.

Kind regards,

Sergio Consoli

Academic Editor

PLOS ONE

Additional Editor Comments (optional):

Reviewers' comments:

Reviewer's Responses to Questions

**Comments to the Author**

1. If the authors have adequately addressed your comments raised in a previous round of review and you feel that this manuscript is now acceptable for publication, you may indicate that here to bypass the “Comments to the Author” section, enter your conflict of interest statement in the “Confidential to Editor” section, and submit your "Accept" recommendation.

Reviewer #1: All comments have been addressed

Reviewer #3: All comments have been addressed

2. Is the manuscript technically sound, and do the data support the conclusions?

Reviewer #1: (No Response)

Reviewer #3: Yes

3. Has the statistical analysis been performed appropriately and rigorously? 

Reviewer #1: (No Response)

Reviewer #3: N/A

4. Have the authors made all data underlying the findings in their manuscript fully available?

Reviewer #1: (No Response)

Reviewer #3: Yes

5. Is the manuscript presented in an intelligible fashion and written in standard English?

Reviewer #1: (No Response)

Reviewer #3: Yes

6. Review Comments to the Author

Reviewer #1: I have read the revised version and found it acceptable for publication. I think the authors achieved introducing a new method to synthesize verbally conceptual models in ecology.

Reviewer #3: I thank the authors for their consideration of my comments in the previous round. I am pleased to see that all comments were addressed suitably. Regarding Point 2, I was concerned about potential bias caused by the makeup of the

community of scientists under study, thus, the comment to Rev2 in the previous round suffices. Well argued. Congratulations on this work. I am happy to endorse its publication. Best wishes

7. PLOS authors have the option to publish the peer review history of their article (what does this mean?). If published, this will include your full peer review and any attached files.

Reviewer #1: No

Reviewer #3: No

---

## [Editor Report · Acceptance letter]

9 Dec 2021

PONE-D-21-03811R3 

A pragmatic approach for producing theoretical syntheses in ecology 

Dear Dr. Travassos-Britto:

I'm pleased to inform you that your manuscript has been deemed suitable for publication in PLOS ONE. Congratulations! Your manuscript is now with our production department. 

Kind regards, 

on behalf of

Dr. Sergio Consoli 

Academic Editor

PLOS ONE